# Revisiting Noise Resilience Strategies in Gesture Recognition: Short-Term Enhancement in sEMG Analysis

**Weiyu Guo** [1]  **Ziyue Qiao** [2]  **Ying Sun** [1 3]  **Yijie Xu** [1]  **Hui Xiong** [1 3]

## Abstract

Gesture recognition based on surface electromyography (sEMG) has been gaining importance in many 3D Interactive Scenes. However, sEMG is easily influenced by various forms of noise in real-world environments, leading to challenges in providing long-term stable interactions through sEMG. Existing methods often struggle to enhance model noise resilience through various predefined data augmentation techniques. In this work, we revisit the problem from a short-term enhancement perspective to improve precision and robustness against various common noisy scenarios with learnable denoise using sEMG intrinsic pattern information and sliding-window attention. We propose the Short Term Enhancement Module(**STEM**), which can be easily integrated with various models. **STEM** offers several benefits: 1) Noise-resistant, enhanced robustness against noise without manual data augmentation; 2) Adaptability, adaptable to various models; and 3) Inference efficiency, achieving short-term enhancement through minimal weight-sharing in an efficient attention mechanism. In particular, we incorporate **STEM** into a transformer, creating the Short-Term Enhanced Transformer (**STET**). Compared with best-competing approaches, the impact of noise on **STET** is reduced by more than 20%. We report promising results on classification and regression tasks and demonstrate that **STEM** generalizes across different gesture recognition tasks. The code is available at https://github.com/guoweiyu/short-term-semg.

[1]Thrust of Artificial Intelligence, The Hong Kong University of Science and Technology (Guangzhou), Guangzhou, China [2]School of Computing and Information Technology, Great Bay University, Dongguan, China [3]Department of Computer Science and Engineering, The Hong Kong University of Science and Technology, Hong Kong SAR, China. Correspondence to: Ying Sun <yings@ust.hk>, Hui Xiong <xionghui@ust.hk>.

*Proceedings of the 42nd International Conference on Machine Learning*, Vancouver, Canada. PMLR 267, 2025. Copyright 2025 by the author(s).

## 1. Introduction

Surface Electromyographic (sEMG) is a non-invasive technique for monitoring muscle neuron firing, which is an effective way to capture human motion intention and has shown great application potential in the field of human-computer interaction (HCI) (Xiong et al., 2021; Liu et al., 2021c; Guo et al., 2024). A schematic diagram of the EMG-based HCI System is shown in Figure 1. Compared to traditional HCI channels, sEMG has the advantages of being generated prior to actual motion (50-150 ms), containing rich motion intention information, and being easy to collect (Sun et al., 2020). Therefore, increasing interest has been in exploring EMG-based motion track (Liu et al., 2021b; Guo et al., 2023) and pathological analysis.

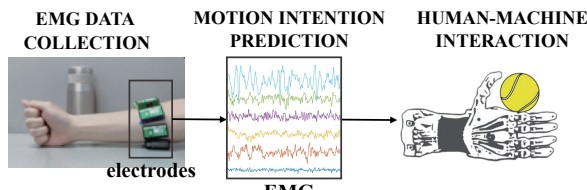

**EMG DATA COLLECTION**    **MOTION INTENTION PREDICTION**    **HUMAN-MACHINE INTERACTION**

electrodes    EMG

Figure 1: Schematic Diagram of EMG-based Human-Computer Interaction System.

Recently, deep learning models for time series have developed rapidly and are widely used (Dai et al.; Zhang et al., 2023a; Xu et al.; Zhang et al., 2021). By treating sEMG as time series, deep sequential models (Bi et al., 2019; Tsinganos et al., 2019; Becker et al., 2018; Li et al., 2021; Du et al., 2017; Zhang et al., 2021) have been applied to sEMG modeling. For example, Zhang et al. (2022) employ a multi-task encoder-decoder framework to improve the robustness of sEMG-based Sign Language Translation (SLT), while Rahimian et al. (2021) employ a Vision Transformer (ViT)-based architecture (TEMGNet) to enhance the accuracy of sEMG-based myocontrol for prosthetics. Although these methods show improved performance over traditional approaches, they process sEMG signals as generic time-series data, without tailoring their designs to the unique characteristics of sEMG—such as high variability and sensitivity to external noise and interference. This oversight leads to challenges in handling low signal-to-noise ratios caused by skin condition changes and interference, and limits the

ability to capture subtle but critical motion information. As a result, the robustness and accuracy of existing models remain significantly constrained.

Processing sEMG signals is challenging due to the complex noise mixed in the skin's surface and the presence of patterns across various time scales. Existing works, mainly focusing on long-term sequences, have used transformers to treat sEMG as a typical time series, aiming to enhance long-term dependencies. These approaches overlook the critical features that are present on short-term scales. Short-time scale features are important in sEMG analysis, as they aid in distinguishing subtle movements and facilitate the removal of variable noise. For example, gestures like Index Finger Extension (IFE) and Middle Extension (ME), while similar in global sEMG patterns, can be differentiated through localized short-term signal variations.

To this end, in this paper, we present a lightweight but powerful module called Short-Term Enhanced Module (**STEM**) which utilizes sliding window attention with weight sharing to capture short-term features. Building on **STEM**, we further propose the Short-Term Enhanced Transformer (**STET**). **STET** leverages **STEM** to capture local signal changes, enhancing noise resistance, and then combines **STEM** with long-term features, further improving accuracy for downstream predictions. Furthermore, to enhance model robustness with minimal annotation, we propose a self-supervised paradigm based on sEMG Signal Masking to leverage the inherent variability in sEMG signals.

Finally, we conducted extensive experiments on the largest public sEMG datasets. The experimental results show that **STET** surpasses existing methods by a significant margin in both gesture classification and joint angle regression tasks for single-finger, multi-finger, wrist, and rest gestures. Meanwhile, **STET** achieves strong robustness even when trained on pure data and tested on noisy data. Compared with best-competing approaches, the impact of noise on **STET** is reduced by more than $20\%$. Moreover, through visualizations, we show that the long-term and short-term features are complementary in sEMG-based gesture recognition tasks, and the fusion of the two features can make the classification boundary more obvious. This clearly demonstrates that short-term information is critical for sEMG-based gesture recognition and will provide a new design paradigm for future sEMG model design. In particular, we have deployed **STET** as an important functional component in our HCI system, which can offer a more intuitive and effective experience. Our real-world deployment is shown in Figure 5 in the appendix.

To the best of our knowledge, we are the first to explicitly emphasize short-term features in sEMG-based gesture recognition. Our main contributions are as follows:

- We propose **STEM**, a learnable, adaptable, and noise-resistant module to enhance short-term features. Integrating **STEM** into various neural network architectures yields consistent and significant performance improvements;
- We introduce sEMG Signal Masking into the self-supervised Intrinsic Pattern Capture module, enabling the model to exploit sEMG variability;
- We conduct comprehensive experiments on the largest available wrist sEMG dataset, demonstrating that our method outperforms existing approaches in both accuracy and robustness. Furthermore, short-term enhancement can generalize to other architectures such as Informer.

## 2. Related Work

**The EMG-based Intention Prediction of Human Motion** can be broadly divided into model-based and data-driven methods. Model-based methods typically combine disciplines such as kinesiology, biomechanics, and human dynamics to explicitly model the relationship between EMG and outputs (such as joint angles and forces). The model often includes specific parameters, such as joint positions and bone-on-bone friction, that need to be repeatedly experimented with and adjusted until the desired performance is achieved. In terms of parameter selection and determination, model-based methods can be further divided into kinematic models (Borbély & Szolgay, 2017), dynamic models (Koike & Kawato, 1995; Koirala et al., 2015; Liu et al., 2015), and muscle-bone models (Wang & Buchanan, 2002; Zhao et al., 2020; Yao et al., 2018). Clancy et al. (2012) used a nonlinear dynamics model to identify the relationship between constant posture electromyography and torque at the elbow joint. Hashemi et al. (2012) used the Parallel Cascade Identification method to establish a mapping between forearm muscles and wrist forces. However, model-based methods have a large number of parameters that are difficult to measure directly. Currently, only simple motion estimation with a limited number of joints and degrees of freedom is possible. In contrast to model-based approaches, data-driven methods do not require the measurement of various parameters. Recently, some researchers have begun to use temporal deep learning models to extract motion information from sEMG (Lin et al., 2022; Zhang et al., 2022; Guo et al., 2021; Lin et al., 2024; Chen et al., 2024; Jiang et al., 2025). Lin et al. (2022) proposed a BERT-based structure to predict hand movement from the Root Mean Square (RMS) feature of the sEMG signal. Rahimian et al. (2021) proposed a novel Vision Transformer (ViT)-based neural network architecture to classify and recognize upper-limb hand gestures from sEMG for use in myocontrol of prostheses. However, these methods have neglected the modeling of short-term dependencies and have not considered the inherent variability in sEMG signals. Furthermore, there is a lack of research in this field aimed at improving noise robustness.

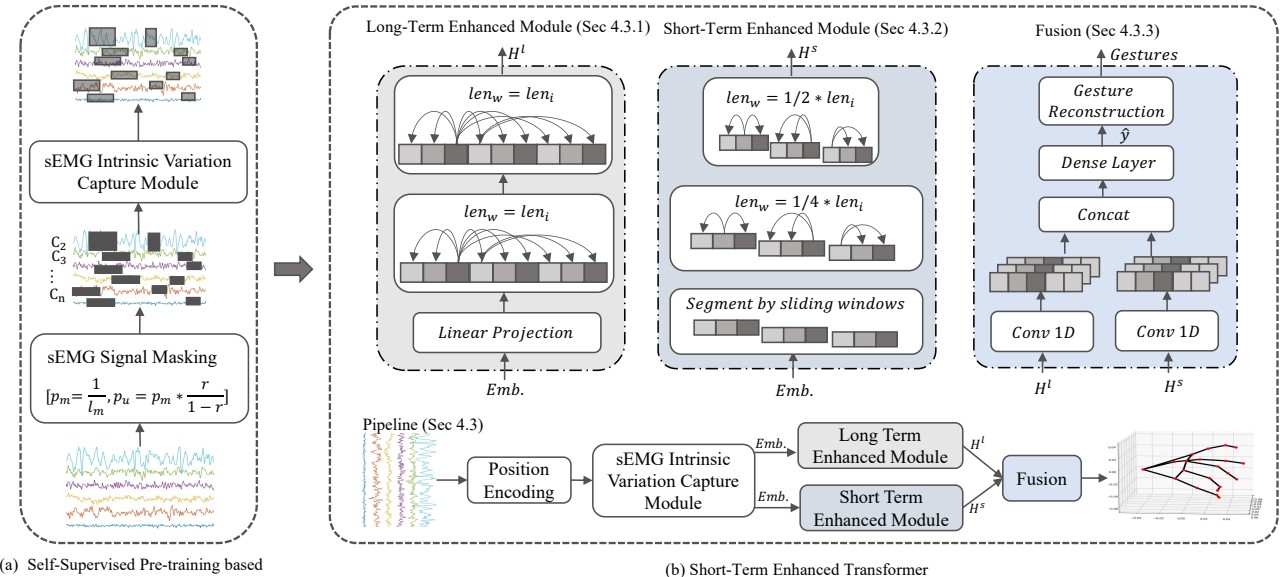

Figure 2: Overview of **STET**. The sEMG signal is encoded using the sEMG Intrinsic Pattern Capture module, which is first pre-trained via sEMG signal Masking. A long-term and short-term enhanced module improves sEMG representations. An asymmetric optimization strategy addresses biases and imbalances in gesture recognition through an asymmetric classification loss.

## 3. Preliminaries

### 3.1. Dataset

We conduct the experiments on the GRABMyo (Pradhan et al., 2022) and the Ninapro DB2 (Atzori et al., 2014) datasets, which are the largest and most widely used sEMG datasets and have great potential for developing new generation human-machine interaction based on sEMG signals.

*Data processing.* The subjects performed 17 gestures of hand and wrist (including a rest period sEMG) according to the prompts on the computer screen. Each gesture was repeated 7 times, each lasting 5 seconds. To avoid muscle fatigue, rest 10s between repetitions. In the following experiments, we use five repetitions as the training set and two repetitions as the test set. Bandpass filtered between 10 Hz and 500 Hz with a gain of 500 was adopted to the raw signal. We use the difference between the corresponding electrodes in the two loops as the input signal for our model. To improve the convergence speed of the model, we use two methods (Max-Min normalization, $\mu$-law normalization) to normalize the data (Rahimian et al., 2020; Recommendation, 1988). After normalization, we use a time-sliding window to split samples. We set the window size as 200ms, and the overlap of adjacent windows is 10ms. $\mu$-law normalization can logarithmically amplify the outputs of sensors with small magnitudes, which results in better performance than linear normalization.

**Definition 3.1** (sEMG Signal Sequence). An sEMG signal sequence is defined as a temporal signal sequence sampled by multiple sensors from a human wrist, which can be formulated as $\mathbf{X} = [\mathbf{x}_1, \mathbf{x}_2, .., \mathbf{x}_t]$, where $t$ is the time window

and $\mathbf{x}_i = [x_{i,1}, x_{i,2}, ...x_{i,c}]$ represents the signal, vector of $c$ sensors, where $x_{i,j}$ is the signal value of the $j$-th sensor in the $i$-th time step.

## 4. Technical Details

### 4.1. Model Overview

Figure 2 illustrates the overview of our proposed framework for gesture recognition, which contains three components: (1) The *sEMG Intrinsic Pattern Capture* module encodes the sEMG signal sequence into the hidden sEMG representations. A pre-training model with a segment masking strategy and MSE reconstructing loss is proposed to learn inherent variability from the sEMG signals into the model's parameters. f (2) The *Long-term and Short-term Enhanced* module uses two decoupling heads to extract the long-term and short-term context information separately, which improves the sEMG representations in preserving both the global sEMG structure and multiple local signal changes of the sEMG. (3) The *Asymmetric Optimization* strategy addresses the problems of sample biases and imbalance in gesture recognition via an asymmetric classification loss, which can make the model focus on hard and positive samples to improve the recognition.

### 4.2. sEMG Intrinsic Pattern Capture Module

#### 4.2.1. SEMG SIGNAL ENCODING

Given the sEMG signal sequence $\mathbf{X} = [\mathbf{x}_1, \mathbf{x}_2, .., \mathbf{x}_t]$, we first project each signal $\mathbf{x}_i \in \mathbb{R}^c$ into a hidden embedding via a transformation matrix and add each signal embedding with an absolute position embedding. Then, we feed the output sequence into a $L$-layer Transformer and obtain

the output signal embeddings $\mathbf{X}^{(L)} = [\mathbf{x}_1^{(L)}, \mathbf{x}_2^{(L)}, .., \mathbf{x}_t^{(L)}]$, which incorporate temporal context signal information for each position in the sequence.

### 4.2.2. sEMG SIGNAL MASKING

After the sEMG signal-extracting module is constructed, we aim to use pre-training to exploit the intrinsic pattern and temporal semantics disclosed by the unlabeled sEMG signals (labeling sEMG is time-consuming and labor-intensive) and give a good initialization for the model parameters, then avoid the model focusing on some noisy features in the supervised learning task to over-fitting on some local minimums. Thus, we propose a sEMG Intrinsic Pattern Capture based on a signal masking strategy detailed in Algorithm 1.

Specifically, given a transformed signal embedding sequence $\mathbf{X} = [\mathbf{x}_1, \mathbf{x}_2, .., \mathbf{x}_t]$, instead of adding masks on the sequence in terms of time steps like BERT, we add sensorwise masks for the signal sequence of each sensor similar with (Zerveas et al., 2021), which can encourage the model to learn more fine-grained temporal context dependency on the signal sequence of multiple electrodes. For the signal sequence of the $i$-th sensor, formulated as $[x_{1,i}, x_{2,i}, ..., x_{t,i}]$, i.e, the $i$-th column of $\mathbf{X}$, we generate a binary mask vector $\mathbf{m}_i \in \mathbb{R}^t$, where average $r$ radio of elements in $\mathbf{m}_i$ should be 0.15. Randomly generating $\mathbf{m}_i$ may cause a lot of isolated-masked signals, meaning one masked signal whose adjacent signals are unmasked. However, a single signal can be easily predicted by its immediately preceding or succeeding signals, making self-supervised learning easy to fit on ineffective patterns and poor for learning temporal semantic information. Considering this, we introduce a more complex masking strategy that aims to generate multiple masked segments on the sequence with an average length $l_m$, which means $m_i$ is composed of contiguous masked segments and unmasked segments. The length of masked segments follows a geometric distribution with mean $l_m$, and the length of unmasked segments follows a geometric distribution with mean $l_u$. Also, $l_m/l_u = r/(1-r)$ so that the number of masked elements would follow the proportion $r$. While Miao et al. (2022) inspired our use of geometric distribution for segment lengths, we extend this concept by applying the masking strategy in a sensor-wise manner and employing different sequence reconstruction methods.

Then, we can mask the input sEMG signal sequence $\mathbf{X}$ by $\widehat{\mathbf{X}} = \mathbf{X} \odot \mathbf{M}$, where $\odot$ is elementwise multiplication and $\widehat{\mathbf{X}}$ is the masked input. With the proposed Transformer-based sEMG signal encoder, we can obtain the output $\widehat{\mathbf{X}}^{(L)} = [\widehat{\mathbf{x}}_1^{(L)}, \widehat{\mathbf{x}}_2^{(L)}, ..., \widehat{\mathbf{x}}_t^{(L)}]$. For self-supervised learning, we add a linear layer on the top of masked output to reconstruct each sEMG signal $\widehat{\mathbf{x}}_i^{(L)}$ as $\widetilde{\mathbf{x}}_i \in \mathbb{R}^c$, the reconstructed sEMG signal in the $i$-th time step generated from the masked input. Then, we minimize the Mean Squared Error (MSE) of the reconstructed signals and original signals on the masked

positions for each sample: The optimization objective $\mathcal{L}_{pt}$ of pre-training can be written as follows:

$$\min \frac{1}{|\mathbf{M}|} \sum_{i=0}^{t} \sum_{j=0}^{c} \mathbb{1}(\mathbf{M}_{i,j} = 0)(\widetilde{x}_{i,j} - x_{i,j})^2, \quad (1)$$

where $\mathbb{1}(\cdot)$ is the indicator function, $\widetilde{x}_{i,j}$ and $x_{i,j}$ are the reconstructed value and original value of $j$-th sensor in $\widetilde{\mathbf{x}}_i$ and $\mathbf{x}_i$ respectively, and $\mathbf{M}_{i,j}$ is the element in the $i$-th row and the $j$-th column of $\mathbf{M}$. Thus, we can pre-train the sEMG Intrinsic Pattern Capture via the above strategy to obtain well-initialized model parameters for the downstream task. In practice, we empirically set the masking proportion $r$ as 0.15 and the average length of masked segments as 3. The pre-training procedure is illustrated in Figure 2 (a).

***Sensor-wise Mask:*** Each electrode has distinct conditions (e.g., moisture level, coating thickness), resulting in varying signal quality. In real-world scenarios, many types of noise are sensor-specific (e.g., signal loss from certain electrodes or poor electrode contact).

***Contiguous Masked Segments:*** The primary frequency range of sEMG signals is 20–500 Hz, while dataset sampling rates are typically around 2000 Hz. This results in similar values for adjacent sampling points. Contiguous masking prevents the model from relying on adjacent sampling points to predict the masked values.

***The Length of Masked Segments Follows a Geometric Distribution:*** The effective frequency range of sEMG signals varies within 20–500 Hz. Frequency changes manifest as variations in the number of effective sampling points in the time domain. Using dynamic-length masked segments better captures these variations. Geometric distribution aligns with the characteristics of electrode distribution and the frequency dynamics of sEMG signals.

### 4.3. Long-term and Short-term Decoding

Based on the pre-trained sEMG Intrinsic Pattern Capture, we develop two decoder heads to extract long-term and short-term dependency on the signal sequences, respectively. Intuitively, the long-term and short-term information on signal sequences is significant in the gesture recognition problem. Long-term information refers to the global context of an sEMG sequence, which provides a signal's overall structure to help interpret the gesture. Short-term information refers to the movement signal in a short time interval of the whole sequence, which can provide specific local characteristics for accurate recognition when the overall structures of sEMGs are ambiguous. For example, distinguishing between Index Finger Extension (IFE) and Middle Extension (ME) movements requires a closer examination of the local signal changes in sEMG, whereas differentiating gestures with large variations, such as hand gestures and wrist gestures, necessitates a focus on the global sEMG information.

### 4.3.1. PRESERVING LONG-TERM SEMG SIGNAL

Given the hidden output $\mathbf{X}^{(L)} = [\mathbf{x}_1^{(L)}, \mathbf{x}_2^{(L)}, ..., \mathbf{x}_t^{(L)}]$ of a sEMG signal sequence, we first build a long-term decoder to extract the long-term dependency on the complete output. Specifically, the long-term decoder is defined as a multi-head self-attention layer.

The global context signal information is collected to the embeddings of $t$ timesteps with different attention weights through the self-attention layer. The detailed equation is shown in Equation 4a in the appendix.

### 4.3.2. PRESERVING SHORT-TERM SEMG SIGNAL

We introduce a slide-window self-attention layer to model the local context information within a short time interval to extract the short-term dependency on the signal outputs. Similarly, we stack multiple attention heads and calculate the attention of context signals to sum them into the final representations. The difference is we only calculate the attention of its nearest $w$ context for each time step. Specifically, we can rewrite the $\text{Attention}(\cdot)$ in Equation 4b as:

$$\text{Attention}_S(\mathbf{Q}, \mathbf{K}, \mathbf{V}) = \left[\text{Softmax}\left(\frac{\mathbf{Q}_i \mathbf{K}_i^{wT}}{\sqrt{h}}\right) \mathbf{V}_i^w\right]_{i=1}^t, \tag{2a}$$

$$[\mathbf{K}_i^w]_{i=1}^t = \text{Unfold}(\mathbf{K}, w), \tag{2b}$$

$$[\mathbf{V}_i^w]_{i=1}^t = \text{Unfold}(\mathbf{V}, w), \tag{2c}$$

where $w$ is the sliding windows size, $\mathbf{Q}_i \in \mathbb{R}^h$ as the $i$-th query is the $i$-th row of $\mathbf{Q}$, $\mathbf{K}_i^w \in \mathbb{R}^{w \times h}$ and $\mathbf{V}_i^w \in \mathbb{R}^{w \times h}$ are the keys and values in a window around the $i$-th query. We utilize the $\text{Unfold}(\cdot)$ operation for the key and value matrix to generate the sliding windows for each timestep. Note that we omit the index of attention head in the above equation to avoid confusion.

Thus, by stacking multiple sets of parameters in $\text{Attention}_S(\cdot)$ to constitute different attention heads, we can obtain the short-term sEMG embeddings $\mathbf{H}^s \in \mathbb{R}^{t \times h}$ by $\mathbf{H}^s = \text{MultiHead}_S(\mathbf{X}^{(L)})$. Using sliding windows, each row in $\mathbf{H}^s$ preserves the local context information of the corresponding timestep, representing the movement from the past $w/2$ timesteps to the next $w/2$ timesteps.

In contrast to Focal Transformer (Yang et al., 2021), which processes long- and short-term features sequentially, we handle them in parallel. While Zhang et al. (2023b) adopts patch-based segmentation, our method uses sliding-window attention to enable fine-grained feature extraction and better sensitivity to boundary variations. Zhu et al. (2021) combine long-range low-rank attention with short-range sliding-window attention via unified key/value sets. In comparison, our model uses two decoupled heads to separately capture long- and short-term dependencies.

### 4.3.3. FUSION

Obtained the long-term embeddings $\mathbf{H}^l$ and the short-term embeddings $\mathbf{H}^s$ of an sEMG signal sequence, we first concatenate them in terms of the hidden dimension, then introduce a 1-D convolution to summarize the $t$-step sEMG embedding sequence into the final sEMG representation, which is fed into a *Feed Forward Layer* with a *Sigmoid Layer* to obtain the final classification probability of which gesture the sEMG belonging to, which can be written as: $\mathbf{h} = \mathbf{u}^T \cdot [\mathbf{H}^l : \mathbf{H}^s]$ and $\widehat{\mathbf{y}} = \sigma(\text{FC}(\mathbf{h}))$, where $\mathbf{h} \in \mathbb{R}^{2h}$ is the final sEMG representation, $\mathbf{u} \in \mathbb{R}^t$ is the weight parameters to weighted sum up the sEMG embeddings in each time steps, $\text{FC}(\cdot)$ is a two-layer fully connected layer, $\sigma(\cdot)$ is the activation function, and $\widehat{\mathbf{y}} \in \mathbb{R}^C$ is the output classification probability of the sEMG signals.

### 4.4. Asymmetric Optimization

Following common practice in multi-label classification, we decompose gesture recognition into a series of binary classification tasks. However, two key challenges arise under this formulation. First, due to temporal instability in the sampled signals, the sEMG sequences may exhibit significant sample-wise bias. Some samples with strong signals are easily predicted, while others with ambiguous or noisy signals are much harder to classify. Second, the sEMG dataset contains 17 gesture classes with balanced sample counts per class. However, for each binary task, the number of negative samples significantly exceeds that of positives. This class imbalance can suppress gradients from positive instances during training, degrading model performance (Liu et al., 2021a).

To address these issues, we adopt the Asymmetric Loss (Ridnik et al., 2021) for gesture classification. This loss, a variant of focal loss, (1) applies focusing parameters to downweight easy samples and emphasize harder ones; and (2) introduces asymmetric focusing and probability shifting to reduce the dominance of abundant easy negatives while preserving the contribution of positives. We define the loss function as follows:

$$\mathcal{L}_{\text{STET}} = -\sum_{i=1}^N \sum_{j=1}^C \left( y_{i,j} \left(1 - \widehat{y}_{i,j}\right)^{\gamma^+} \log\left(\widehat{y}_{i,j}\right) \right.$$
$$\left. + \left(1 - y_{i,j}\right) \left(\widehat{y}_{i,j}^m\right)^{\gamma^-} \log\left(1 - \widehat{y}_{i,j}^m\right), \right. \tag{3}$$

and $\widehat{y}_{i,j}^m = \max\left(\widehat{y}_{i,j} - m, 0\right)$, where $y_{i,j}$ and $\widehat{y}_{i,j}$ is the ground-truth and probability of the $i$-th sEMG signal sequence belonging to the $j$-th gesture. $\left(1 - \widehat{y}_{i,j}\right)^{\gamma^+}$ and $\left(\widehat{y}_{i,j}^m\right)^{\gamma^-}$ are two terms to make the weights of hard predicted samples bigger than those easily predicted samples, $\gamma^+$, and $\gamma^-$ are two focusing parameters and $\gamma^+ > \gamma^-$ lead to asymmetric focusing that help the optimization pay more focus on positive samples of each class. $\widehat{y}_{i,j}^m$ is the shifted

Table 1: The table presents the accuracy (**ACC**, %) and standard deviation (**STD**) of various models evaluated on Single-finger, Multi-finger, Wrist, Rest, and Overall gesture categories. The proposed **STET** model shows the highest performance across all categories.

| Model | Single-finger | | Multi-finger | | Wrist | | Rest | | Overall | |
|---|---|---|---|---|---|---|---|---|---|---|
| | ACC | STD | ACC | STD | ACC | STD | ACC | STD | ACC | STD |
| Asif *et al.* (Asif et al., 2020) | 83.44 | 0.02 | 83.58 | 0.01 | 89.40 | 0.01 | 90.86 | 0.01 | 85.34 | 0.01 |
| TCN (Tsinganos et al., 2019) | 78.78 | 0.02 | 79.10 | 0.02 | 87.27 | 0.01 | 88.57 | 0.02 | 81.50 | 0.02 |
| GRU (Chen et al., 2021) | 84.45 | 0.02 | 84.88 | 0.01 | 90.06 | 0.01 | 89.42 | 0.02 | 86.30 | 0.02 |
| TEMGNet (Rahimian et al., 2021) | 77.70 | 0.02 | 74.00 | 0.03 | 84.04 | 0.02 | 87.46 | 0.01 | 78.02 | 0.02 |
| Zerveas *et al.* (Zerveas et al., 2021) | 78.45 | 0.02 | 77.20 | 0.02 | 87.28 | 0.02 | 86.76 | 0.02 | 80.43 | 0.02 |
| Informer (Zhou et al., 2021) | 86.88 | 0.02 | 86.54 | 0.02 | 91.90 | 0.01 | 83.56 | 0.02 | 87.71 | 0.02 |
| LST-EMG-Net (Zhang et al., 2023b) | 87.21 | 0.01 | 83.16 | 0.01 | 88.36 | 0.02 | 82.52 | 0.02 | 85.31 | 0.02 |
| TEMGNet+**STEM** (ours) | 84.57 | 0.02 | 81.23 | 0.02 | 88.12 | 0.02 | 88.74 | 0.01 | 84.07 | 0.02 |
| Informer+**STEM** (ours) | 87.42 | 0.02 | 88.39 | 0.02 | 92.07 | 0.01 | 90.33 | 0.01 | 89.14 | 0.02 |
| **STET (ours)** | **88.27** | 0.01 | **89.93** | 0.02 | **93.77** | 0.01 | **95.33** | 0.01 | **90.76** | 0.01 |

probability and $m$ is shifting margin. The probability shifting for negative samples encourages the optimizer to further reduce their contribution.

## 5. Experiment

### 5.1. Settings

**Implementation Details** **STET** is implemented in PyTorch (Paszke et al., 2019) and is trained using one RTX 3090 GPU. During training, we use the RAdam (Liu et al., 2019), which is a theoretically sound variant of the Adam optimizer with a weight decay of 1e-3. For classification tasks, we conduct user-specific pretraining on the GRABMyo dataset, and for regression tasks, we carry out user-specific pretraining on the NinaPro DB2 dataset. In the decoder, we use two layers of full attention in the long-term decoder and two layers of sliding window attention in the short-term decoder. The short-term decoder's window size is 41 and 21, and the window's move step is set to 1. In both the pre-training and fine-tuning periods, we set the batch size to 16 and set drop out to 0.2.

#### 5.1.1. EVALUATION METRICS

Following the prior works (Guo et al., 2021; Wang et al., 2020a; Chen et al., 2021; Rahimian et al., 2021), we choose the below metrics to evaluate the model's performance:

**Pearson Correlation Coefficient** (CC) is a widely used measure of the linear relationship between two variables. It ranges from -1 to 1, where a larger CC value indicates greater similarity between the predicted and estimated joint angles curve, indicating improved estimation.

**Root Mean Square Error** (RMSE) is a common metric for evaluating the deviation between predicted and observed values. As the range of fluctuations in the curves of different joint angles can vary significantly, it is difficult to fairly evaluate the performance of models using RMSE alone. Normalization of RMSE addresses this issue, resulting in the Normalized RMSE (NRMSE).

**Average curvature** ($\kappa$) of all points for each joint is used to measure the smoothness of an estimated curve. A smaller $\kappa$ indicates a smoother curve. Details above are covered in appendix Appendix G.3.

### 5.2. Comparison with Baselines

We evaluate the accuracy (ACC) and standard deviation (STD) of our proposed **STET** in comparison with existing sEMG-based gesture recognition methods. Specifically, we train the model on the GRABMyo dataset (Pradhan et al., 2022), with data processing details provided in Section 3.1. Classification results are separately reported for single-finger gestures, multi-finger gestures, wrist gestures, rest, and the overall performance. The ratio of the training set to the test set for each gesture is 5 to 2. The experiment results are shown in Table 1.

The experimental results show that **STET** consistently achieves the best performance across all four gesture categories as well as the overall dataset. In particular, **STET** and Zerveas et al. (2021) both employ Transformer-based encoders. However, in the decoder stage, **STET** integrates both short-term and long-term decoders, in contrast to the fully connected layers used in Zerveas et al. (2021). As a result, the overall accuracy is improved from 80.43% to 90.76%. This improvement stems from the proposed long-term and short-term decoupling module, which captures both global and fine-grained dependencies in sEMG signals, leading to more informative representations.

Among the transformer-based methods, Informer and **STET** performed best, with accuracy rates of 87.71% and 90.76%, respectively. Informer relies heavily on max pooling layers to aggregate features, leading to the relative weakness in extracting some short-term features. **STET** enhances accuracy and stability by strengthening the short-term feature extraction. The improvement is remarkable on the Rest gestures, where the accuracy improves from 83.56% to 95.33%. Furthermore, after incorporating our designed short-term encoder into Informer, its accuracy rate increased from 87.71%

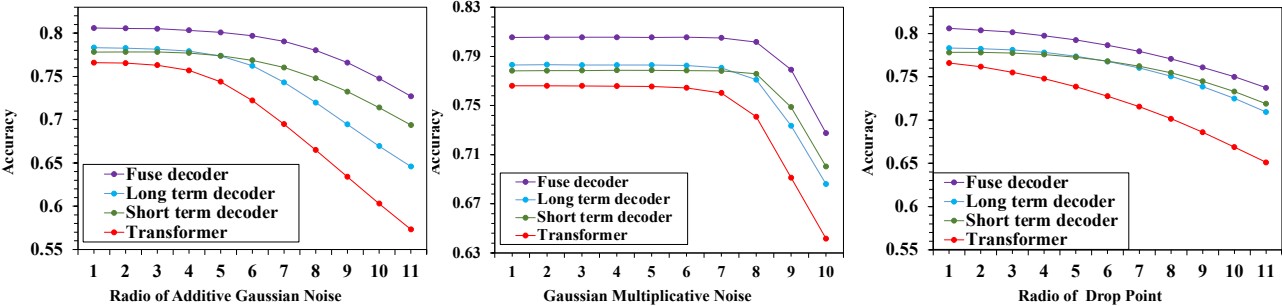

Figure 3: Accuracy versus Noise Intensity Curve. The accuracy of different decoders (Fuse, Long term, Short term, and Transformer) under varying levels of Additive Gaussian Noise, Gaussian Multiplicative Noise, and Drop Point Noise.

Table 2: Ablation study results. **EIPC**: sEMG Intrinsic Pattern Capture; **LT**: Long-Term decoder; **ST**: Short-Term decoder; **CEL**: Cross Entropy Loss; **ASL**: Asymmetric Loss.

| Transformer | EIPC | LT | ST | Fusing | CEL | ASL | ACC (%) |
|:---:|:---:|:---:|:---:|:---:|:---:|:---:|:---:|
| ✓ | | | | | ✓ | | 85.73 |
| ✓ | ✓ | | | | ✓ | | 86.33 |
| ✓ | ✓ | ✓ | | | ✓ | | 88.02 |
| ✓ | ✓ | | ✓ | | ✓ | | 87.72 |
| ✓ | ✓ | ✓ | ✓ | ✓ | ✓ | | 89.37 |
| ✓ | | ✓ | ✓ | ✓ | | ✓ | 89.42 |
| ✓ | ✓ | ✓ | ✓ | ✓ | | ✓ | **90.54** |

to 89.14%, and the classification accuracy for Rest gestures improved from 83.56% to 90.33%.

The *Rest* gesture, representing the idle state of devices such as interactive bracelets, is the most frequent gesture class in the dataset. As a result, the stability of its prediction is crucial for reliable gesture recognition. **STET** achieves the highest and most stable performance on the *Rest* category compared to all baseline methods, demonstrating its robustness in handling frequent and critical gesture states.

### 5.3. Ablation Studies

To validate the effects of the unsupervised sEMG Intrinsic Pattern Capture (EIPC), Long-term decoder, Short-term decoder, Fuse strategy, and loss function. We designed variants of **STET** and reported their results in Table 2.

First, we observe that with the introduction of unsupervised EIPC, the accuracy of both the Transformer and **STET** improves by 0.60% and 1.12%, respectively, compared to training from scratch. This suggests that EIPC enables the model to capture additional features from the data—such as the inherent variability in sEMG—without the need for new samples or additional annotations. Importantly, this process avoids reliance on external data, thereby preserving user privacy during data acquisition and processing. Replacing the fully connected layer in the Transformer's decoder with either the long-term decoder or the short-term decoder improves performance by 1.16% and 1.39%, respectively. The similar effectiveness of both variants indicates that long-term and short-term features play distinct and complemen-

tary roles in sEMG signal recognition, and that short-term information should not be neglected. Most notably, when our fusion module is used to combine long-term and short-term features, accuracy further increases by 2.46%. This result confirms that the two decoders are complementary and that explicitly enhancing short-term features on top of long-term modeling is beneficial. Finally, we apply Asymmetric Loss (ASL) to emphasize difficult samples during training. Compared to using Cross-Entropy Loss (CEL), this leads to a 0.73% improvement in accuracy, demonstrating the effectiveness of ASL in improving model robustness.

### 5.4. Robustness Analysis

To verify the model's robustness, we only used high-quality data collected in the lab to train the model and added different types of noise (Additive and Multiplicative Gaussian noise and signal loss) during validation to simulate complex scenarios that might be encountered in real situations.

Additive noise typically refers to thermal noise added to the original signal. This type of noise exists regardless of the presence of the original signal and is often considered the background noise of the system in sEMG acquisition. Multiplicative noise is generally caused by channel instability and has a multiplicative relationship with the original signal. Also, we simulated signal loss during transmission by randomly setting a portion of the signals to zero.

Figure 3 illustrates the influence exerted by three distinctive noise categories, namely additive noise, multiplicative noise, and signal loss, on the accuracy of the proposed model. The model using only the short-term decoder is less affected by noise than the long-term version. This relative robustness of the short-term decoder is potentially attributable to its unique capability to mitigate the global impact of noise by virtue of a sliding window multiple-sampling scheme, which effectively confines the sphere of noise impact. The model that integrates both long-term and short-term characteristics persistently outperforms models that rely on only one. This highlights the significant effectiveness of the integrating process in dealing with noise-induced interference.

As depicted in Table 3, it is evident that both Transformer and Informer models demonstrate a notable enhancement

Table 3: Drop rates of accuracy calculated by $\text{drop rate} = (\text{ACC}_{\text{raw}} - \text{ACC}_{\text{noise}})/\text{ACC}_{\text{raw}}$. **AG**: Additive Gaussian noise, **MG**: Multiplicative Gaussian noise.

| Backbone | In STET | AG Noise | MG Noise | Signal Loss |
|---|---|---|---|---|
| Transformer | No | 25% | 16% | 14% |
| Transformer | Yes | 10% | 10% | 8% |
| Informer | No | 11% | 9% | 26% |
| Informer | Yes | 9% | 8% | 17% |

Table 4: Model performance comparison. **PCC**: Pearson Correlation Coefficient, **NRMSE**: Normalized Root Mean Squared Error.

| Model | PCC ↑ | NRMSE ↓ | $\kappa$ | Epoch Time (s) ↓ |
|---|---|---|---|---|
| LSTM | 0.779 | 0.096 | 0.581 | 26.36 |
| TCN | 0.833 | 0.088 | 1.533 | **3.62** |
| BERT | 0.867 | 0.077 | 1.571 | 4.95 |
| sBERT-OHME | 0.869 | 0.076 | 0.532 | 4.96 |
| **STET** (Ours) | **0.877** | **0.073** | 0.522 | 6.83 |

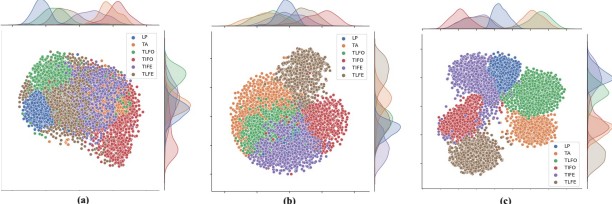

Figure 4: Visualization of (a) the long-term sEMG embeddings, (b) the short-term sEMG embeddings, and (c) the fused sEMG embeddings for gesture recognition. Note that we color each sample by its classes.

in noise resistance when they adopt the design from **STET**. Our short-term enhancement can be easily extended to other models. It is evident that both Transformer and Informer models demonstrate a notable enhancement in noise resistance when their decoders are replaced with the design from **STET**. Specifically, when comparing the drop rates of accuracy due to different noise types, calculated by drop rate:

**Transformer**: Without the **STET** framework, it experienced a drop of 25% under additive Gaussian noise, 16% under multiplicative Gaussian noise, and 14% with signal loss. However, when integrated into the **STET** framework, these drops were reduced to 10% for both additive and multiplicative Gaussian noise and 8% for signal loss.

**Informer**: Without the **STET** framework, it showed a drop of 11% due to additive Gaussian noise, 9% due to multiplicative Gaussian noise, and a significant 26% with signal loss. With the **STET** design, these rates improved to 9% for additive Gaussian noise, 8% for multiplicative Gaussian noise, and 17% for signal loss.

### 5.5. Visualizations

To demonstrate the distinction of different gestures, we first obtained **STET**'s long-term, short-term, and fuse embeddings. The embeddings with dimensions $(N, T, H)$ were then flattened to $(N, T * H)$ and separately projected in 2D by t-SNE, the result shown in Figure 4. We colored each node by category for the illustration. As shown in Figure 4, The classification boundary generated by the long-term feature and the short-term feature is a significant difference, indicating that the long-term and short-term features are capable of recognizing different types of gestures. This further suggests that the two features are complementary in data representation. For example, short-term embedding can distinguish TA gestures from TIFO gestures very well, but TA gestures and TIFO gestures will be confused in long-

term embedding. Meanwhile, long-term embedding can distinguish LP and TIFE gestures very well, but short-term embedding will confuse them. As shown in Figure 4(c), after the fusion of the two types of features, the classification interface is wider, and the confusion points are significantly reduced, which indicates that the fusion module can effectively complement the strengths of the two types of features.

### 5.6. Regression: Hand Joint Angles Prediction

**STET** can conveniently handle regression tasks by changing the loss function to mean squared error (MSE) loss. Continuous motion estimation extracts continuous motion information, such as joint angles and torques, from sEMG signals. Since continuous motion estimation requires outputting subtle movement variations at each time instant, the local signal variations are particularly important for this type of estimation. In this section, we have re-selected the most competitive models known for sEMG-based joint angle prediction as the baseline and tested the performance of **STET** on the regression task of predicting the main 10 joint angles for fingers using the Ninapro DB2 (Atzori et al., 2014) dataset. As shown in Table 4, **STET** achieved the best performance in PCC, NRMSE, and $\kappa$, indicating that the joint angle curve predicted by **STET** is more in line with the real curve and has less abnormal fluctuations, which will significantly improve the user's interactive experience. In terms of training time, due to the addition of the short-term decoder, its training speed is slightly slower than BERT but still within an acceptable range.

## 6. Conclusion

Current sEMG-based gesture recognition models usually fail to handle various noisy and distinguish similar gestures, especially in non-laboratory settings. In this paper, we found using short-term information and self-supervised EIPC mitigates this issue. Therefore, we proposed **STEM** to capture local signal changes and enhance noise resistance. **STEM** is easily deployable and serves as a plug-in that can potentially be applied to most time series deep learning models. According to our experimental results, our method significantly improved performance for both classification and regression tasks in sEMG, and the model's ability to resist signal loss, Gaussian additive noise, and Gaussian multiplicative noise was clearly improved. This will further drive the practical application of sEMG in VR, AR, and other human-computer interaction scenarios.

## Software

The code is available at https://github.com/guoweiyu/short-term-semg.

## Acknowledgements

This work was supported in part by the National Key Research and Development Program of China (Grant No.2023YFF0725001), in part by the National Natural Science Foundation of China (Grant No. 92370204, 62306255), in part by the Guangdong Basic and Applied Basic Research Foundation (Grant No.2023B1515120057, 2024A1515011839), in part by Guangzhou-HKUST(GZ) Joint Funding Program (Grant No.2023A03J0008), the Fundamental Research Project of Guangzhou (No. 2024A04J4233), Education Bureau of Guangzhou Municipality.

## Impact Statement

This paper presents work whose goal is to advance the field of Machine Learning, particularly in sEMG-based gesture recognition. We believe our work has several potential positive societal consequences, such as enabling more intuitive and robust control in human-computer interaction, especially for VR/AR applications, and improving assistive technologies like prosthetic hands through more stable and precise motion intention prediction. By enhancing noise resilience, this research can lead to wider adoption and more practical applications of sEMG-based systems in real-world environments, facilitating new developments in fields requiring fine-grained motion analysis and control.

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

# Part I

# Appendix

## A. The details of methods

### A.1. The details of sEMG Signal Masking

The algorithm of sEMG Signal Masking is shown in 1. The algorithm generates a mask matrix $\mathbf{M}$, where each element $\mathbf{m}_{i,j}$ indicates whether the $j$-th sampling point of the $i$-th sensor is masked.

---

**Algorithm 1** The Algorithm of sEMG Signal Masking

---

**Require:** The length of the input signal sequence $t$
**Require:** The number of signal sensors $c$
**Require:** The average length of masked segments $l_m$
**Require:** The masked ratio $r$
**Ensure:** The mask matrix $\mathbf{M}$
 1: **for** $i = 1, \ldots, c$ **do**
 2:    $\mathbf{m}_i \leftarrow [\text{True}]^t$ % Initialize all elements to True
 3:    $p_m \leftarrow \frac{1}{l_m}$ % Probability of ending a masked segment
 4:    $p_u \leftarrow \frac{p_m \cdot r}{1-r}$   % Probability of ending an unmasked segment
 5:    $\mathbf{p} \leftarrow [p_m, p_u]$
 6:    state $\leftarrow \text{Bool}(\text{random}(0,1) > r)$ % Initialize first state
 7:    **for** $j = 1, \ldots, t$ **do**
 8:       $\mathbf{m}_{i,j} \leftarrow$ state
 9:       **if** $\text{random}(0,1) < \mathbf{p}[\text{state}]$ **then**
10:          state $\leftarrow \neg$state {Flip the state}
11:       **end if**
12:    **end for**
13: **end for**
14: **Return** $\mathbf{M} \leftarrow \left([\mathbf{m}_i]_{i=0}^c\right)^T$

---

The choice of a sensor-wise masking strategy aligns well with the practical scenarios of sEMG noise, where signal disturbances often occur at the level of individual sensors. For instance, signal loss from a specific electrode or cases where electrodes become detached or misaligned frequently affects specific sensor data. By employing sensor-wise masking, our approach effectively mimics these real-world noise conditions. Furthermore, it encompasses scenarios of multi-sensor signal loss, broadening the model's generalizability across diverse noise conditions. This strategy encourages the model to focus on learning the intrinsic patterns of sEMG signals from unmasked sensors, enhancing its ability to infer missing data. Unlike other masking techniques, sensor-wise masking does not overly depend on spatial-temporal uniformity across sensors, which is crucial given the localized nature of many sEMG disturbances. This robustness ensures better generalization across various signal conditions, as demonstrated by our model's improved performance in handling noisy and partial signals. By addressing practical noise issues systematically, the proposed approach enhances the model's resilience in real-world applications.

### A.2. Role of unsupervised pretraining and noise robustness

Our sEMG Signal Masking strategy aims to capture intrinsic variability in the signal rather than explicitly removing noise. This improves the model's robustness by:
- Preventing overfitting on spurious patterns in supervised training.
- Enabling better feature learning from unlabeled data.

As shown in Table 3, this approach reduced performance degradation under various noise conditions (e.g., additive Gaussian noise impact decreased from 25% to 10%).To further understand the contribution of the pretrain with sEMG Signal Masking strategy, we conducted additional experiments to evaluate its impact on the model's robustness to various noise types. Specifically, we removed the short-term and long-term enhancement modules and assessed the backbone model (Transformer) under conditions with and without pretraining using the sEMG Signal Masking strategy. Other experimental

conditions, such as ASL loss, remain unchanged. Due to time constraints, this experiment was conducted on data from only three subjects. The results are summarized below:

### A.3. The details of experiment on Table 2

In this experiment, we utilized transformers and informers as the backbone network, employing the same parameters as in Table 1. The **STET** framework incorporates a pretraining strategy with sEMG masks and **STEM** on top of the backbone. In contrast, the No **STET** framework uses a standard pretraining with a general mask, employing a standard masking approach with fixed segment lengths of 3 (aligning with the average length in our method), sensor-agnostic masking, and a random masking ratio of 15% (consistent with our method). Other experimental conditions, such as ASL loss, remain unchanged.

| Backbone | Pretrain with sEMG Signal Masking | AG Noise (%) | MG Noise (%) |
|---|---|---|---|
| Transformer | No | 22.00 | 15.00 |
| Transformer | Yes | 15.00 | 13.00 |

Table 5: Comparison of performance with and without sEMG Signal Masking strategy for Transformer backbone.

### A.4. The definition of long-term encoder

The long-term encoder is defined as follows:

$$\text{MultiHead}_L\left(\mathbf{X}^{(L)}\right) = \text{Concat}\left(h_1, \ldots, h_d\right)\mathbf{W}^O, \tag{4a}$$

$$\text{where} \qquad \{h_i\}_{i=0}^d = \{\text{Attention}\left(\mathbf{Q}_i, \mathbf{K}_i, \mathbf{V}_i\right)\}_{i=0}^d, \tag{4b}$$

$$\text{Attention}\left(\mathbf{Q}_i, \mathbf{K}_i, \mathbf{V}_i\right) = \text{Softmax}\left(\frac{\mathbf{Q}_i\mathbf{K}_i^T}{\sqrt{h}}\right)\mathbf{V}_i, \tag{4c}$$

where $\mathbf{Q}_i = \mathbf{X}^{(L)}\mathbf{W}_i^Q, \mathbf{K}_i = \mathbf{X}^{(L)}\mathbf{W}_i^K, \mathbf{V}_i = \mathbf{X}^{(L)}\mathbf{W}_i^V$ and $\{\mathbf{W}_i^Q, \mathbf{W}_i^K, \mathbf{W}_i^V\}_{i=0}^d \in \mathbb{R}^{h \times h}$ are parameter matrices and $d$ is the number of attention heads. $\text{Concat}(\cdot)$ represents the concatenate operation. $\mathbf{W}^O \in \mathbb{R}^{dh \times h}$ is the output parameter matrix to transform the concatenated outputs of $d$ attention heads. Then, the long-term sEMG embeddings $\mathbf{H}^l \in \mathbb{R}^{t \times h}$ is obtained by $\mathbf{H}^l = \text{MultiHead}_L(\mathbf{X}^{(L)})$.

The linear projection is applied in the long-term enhanced module because this module processes inputs of fixed length due to its structural design. We use linear projection here to adjust the feature dimensions appropriately. In contrast, the short-term enhanced module operates based on a sliding window, which means the input length remains consistent and does not require dimensional adjustment. Therefore, linear projection is not needed in the short-term module.

## B. The significance of the results

We conducted Friedman and Wilcoxon signed-rank tests to analyze significant differences among subjects with different evaluation methods, and we corrected the P-value using Bonferroni correction. Our model outperformed the other models in Table 1 ($P < 0.01$). Finally, we would like to emphasize that in sEMG-related HCI applications, even a small improvement in accuracy can significantly improve user experience. Therefore, we believe that our proposed model can significantly improve user experience.

### B.1. The details of noise

Additive noise typically refers to thermal noise added to the original signal. This type of noise exists regardless of the presence of the original signal and is often considered the background noise of the system in sEMG acquisition. Additive noise can be described as:

$$G_{add}(x) = x + \alpha \cdot N(x), \tag{5}$$

$$N(x) = \frac{1}{\sqrt{2\pi}}\exp\left(-\frac{(x-u)^2}{2\sigma^2}\right), \tag{6}$$

where $G(x)_{add}$ represent the signal with additive noise, $\alpha$ is used to adjust the size of the noise. $N(x)$ is a normal distribution that simulates background noise. Here, we set $u$ to 0 and $\sigma$ to 1.

Multiplicative noise is generally caused by channel instability and has a multiplicative relationship with the original signal. Multiplicative noise used in the experiment can be described as:

$$G_{mul}(x) = x + N_{mul}(x), \tag{7}$$

$$N_{mul}(x) = n \cdot \frac{x^2}{N(x)^2} \setminus 10^{\frac{SNR}{10}}, \tag{8}$$

where $G(x)_{add}$ represent the signal with multiplicative noise, $SNR$ stand for the signal to noise ratio.

Additionally, we simulated signal loss during transmission by randomly setting a portion of the signals to zero.

## C. Real-World Deployment and human-subject study

Although most previous works, including ours, have primarily tested the performance of gesture detection algorithms on offline laboratory datasets (Wang et al., 2020a; Chen et al., 2021), we recognize the importance of real-world application scenarios. These scenarios often involve complex variables such as electrode movement and muscle state changes, which offline testing does not capture adequately. As shown in Figure 1, we expanded our evaluation to include online performance verification to address this gap. This was achieved by integrating our algorithm with a 3D virtual hand, which was developed using the Unreal 5 engine and controlled through **STET** decoding of sEMG signals.

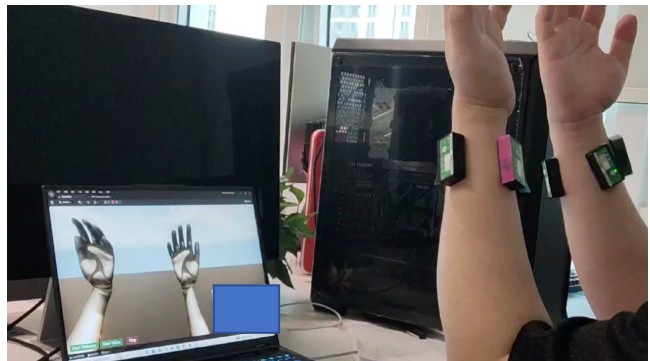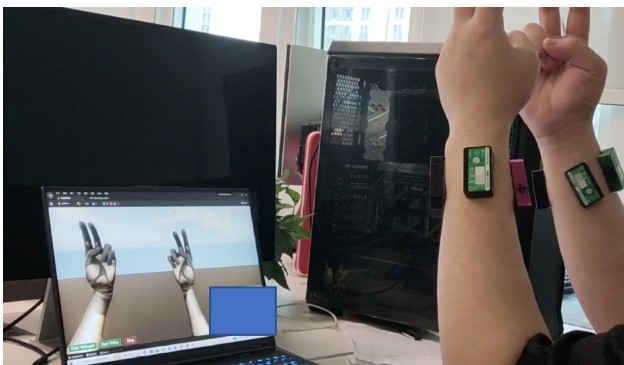
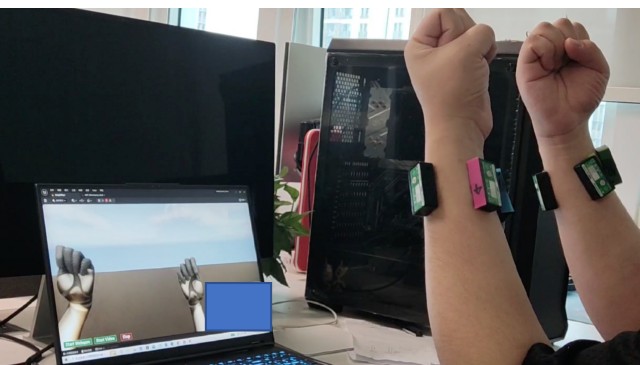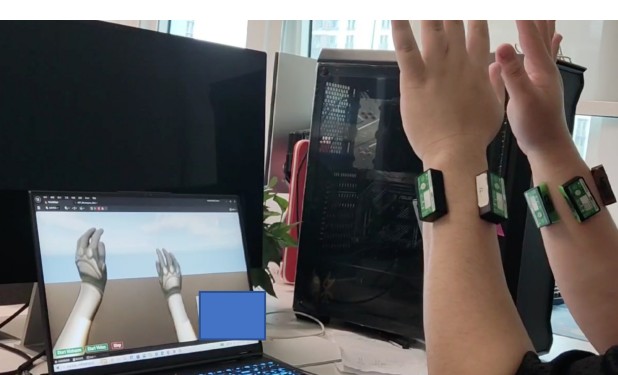

Figure 5: **STET**-based Real-Time Hand Interaction Reconstruction from Wrist sEMG Signals.

To make our system more aligned with daily usage habits, we placed four myoelectric electrodes on both the left and right wrists of users, each with a sampling rate of 2000Hz. The signals were transmitted to the host computer via a WiFi interface for continuous motion estimation. Our online experiment results demonstrated an overall latency of less than 50ms and a Pearson Correlation Coefficient (PCC) of over 0.8. This indicates a stable, accurate, and natural interactive experience, surpassing the capabilities of computer vision-based methods regarding energy efficiency and independence from lighting conditions and occlusions.

To enable streaming processing online, we implemented an online buffer mechanism. Specifically, the buffer collects incoming signals in real-time, and every 10ms, the model retrieves the most recent 200ms of signals from the buffer for processing. The architecture used for online testing is identical to the one described in the paper. Furthermore, the model

has been deployed in a commercial sEMG interaction application, where it has undergone testing in VR control scenarios, demonstrating its practical usability.

In our human-subject study, we recruited a total of eight healthy participants, comprising an equal gender distribution of four males and four females, all of whom were right-handed. Each participant was asked to perform six distinct movements, captured using a standard transformer model and our **STET** model. Importantly, the participants were blinded to the model used during their tasks. After engaging in a 10-minute gaming session designed to test the models, participants were asked to identify which model they felt was more stable and provided a better experience. Remarkably, out of the eight participants, seven preferred the **STET** model, citing its greater stability and overall performance. This overwhelming preference for the **STET** model amongst participants highlights its efficacy and potential for real-world applications, reinforcing our findings regarding its superior performance compared to traditional models.

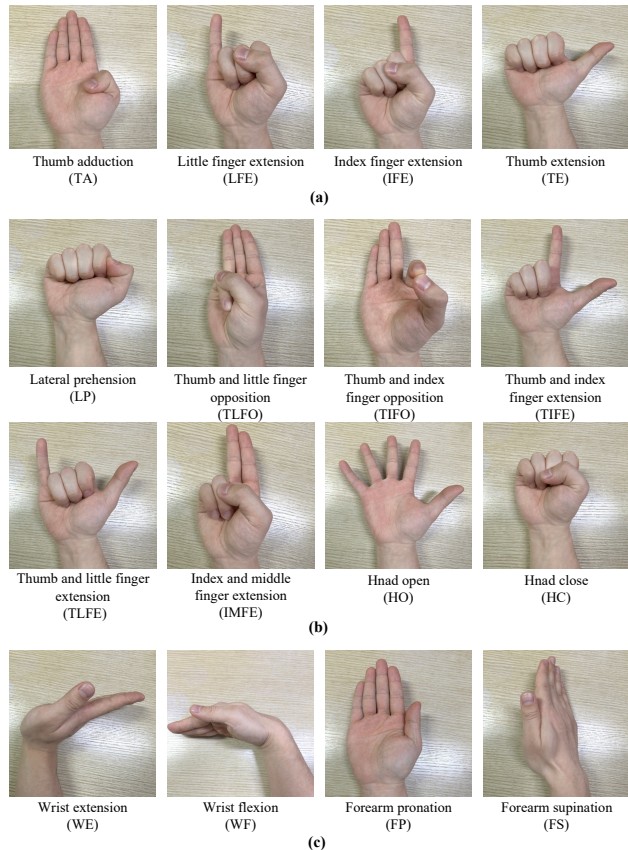

Figure 6: The gestures were used in gesture classification experiment:(a) Single-finger gestures (b) Multi-finger gestures (c) Wrist gestures.

## D. The difference from LST-EMG-NET

Our proposed approach exhibits a marked deviation from the LST-EMG-Net as detailed in (Zhang et al., 2023b). In the LST-EMG-Net, the raw sEMG data is pre-segmented into long and short durations prior to the network ingestion. This pre-segmentation could potentially hinder the encoder's capability to grasp the intrinsic patterns of sEMG. In contrast, our method divides the data into short-term and long-term segments during decoding, which we believe is a more effective strategy. Our results support this, with our method achieving an accuracy of 90.8% compared to 85.3% achieved by the LST-EMG-Net model.

## E. Noise Analysis Metrics

We have employed two key metrics to analyze the noise distribution quantitatively:

**Signal-to-Noise Ratio (SNR)**   We analyzed the spectral properties of each sEMG signal recording and measured the SNR (in dB) as the ratio between the power of the signal to the power of the noise [1]. The power of the noise was estimated as the power of sEMG recordings during the rest trial (when sEMG acquisition is least susceptible to interference) [1]. The average SNR across all signals in our datasets was $14.565 \pm 6.385$ dB.

**Correlation Coefficient of Normality (CCN)**   This metric was used to analyze amplitude distribution. For a static contraction with moderate force, sEMG can be modeled as a filtered, random, white Gaussian noise process [2]. It has been suggested that a test of normality can provide a measure of biosignal quality, where a signal amplitude with a non-Gaussian distribution would be considered contaminated. We generated a Gaussian distribution with equal mean and variance to that of the recording [3]. The CCN is defined as the Pearson correlation coefficient between the histogram bin values of the sEMG recording and the normal density function value for the corresponding bins [4]. A value close to 1 indicates a normal distribution. The CCN of all signals in our dataset was $0.975 \pm 0.041$.

In the real-world dataset, noises are complexly and randomly intermingled, making precise quantitative isolation impossible. Therefore, we used SNR and CCN calculations to analyze the overall noise distribution in spectral and amplitude properties. Our measurements of the GrabMyo dataset revealed the following distribution:

Table 6: SNR Distribution in the GrabMyo Dataset

| Metric | High noise (SNR $< 10$ dB) | Moderate noise ($10$ dB $\leq$ SNR $\leq 20$ dB) | Low noise (SNR $> 22$ dB) |
|---|---|---|---|
| SNR | 28.2% | 56.3% | 15.5% |

Through visual inspection and based on our experience, we identified samples with CCN $< 0.93$ (11.6% of the total samples) as having severe non-Gaussian signals, which may be affected by transient noise such as motion artifacts or electromagnetic interference.

## F. Dataset

We conducted the experiments on the Gesture Recognition and Biometrics ElectroMyogram (GRABMyo) Dataset, the largest known open-source wrist EMG dataset with great potential for developing new generation human-machine interaction based on sEMG. GRABMyo has 43 healthy subjects whose average age is $26.35 \pm 2.89$, and the average forearm length is $25.15 \pm 1.74$ cm (measured as the distance between the olecranon process and the ulnar styloid process). There were 23 male subjects and 20 female subjects, respectively.

*Data collection.* Place 2 rings of 12 monopolar sEMG electrodes (AM-N00S/E, Ambu, Denmark) at the wrist position; each ring consists of 6 electrodes. The sample rate was set to 2048Hz. The distance between the center lines of adjacent electrodes is 2cm. To keep the electrode positions consistent across subjects, the position of the first electrode was fixed at the centerline of the elbow crease. The subjects performed 17 gestures of hand and wrist (including a rest period sEMG) according to the prompts on the computer screen. Each gesture was repeated seven times, each lasting 5 seconds. To avoid muscle fatigue, rest 10s between repetitions. In the following experiments, we use five repetitions as the training set and two repetitions as the test set.

*Data processing.* To avoid muscle fatigue, rest 10s between repetitions. In the following experiments, we use five repetitions as the training set and two repetitions as the test set. Bandpass filtered between 10 Hz and 500 Hz with a gain of 500 was adopted to the raw signal. We use the difference between the corresponding electrodes in the two loops as the input signal of our model.

*Sizes of Training, Validation, and Test Sets.* Table 7 shows the division of the training set, validation set, and test set, which aligns with the common comparison practices in the field. Note that after selecting the parameters using the validation set, the validation set will be merged into the training set for retraining.

**Gesture Classification Task:** For this task, we used the **GRABMyo** dataset for both pre-training and fine-tuning. The GRABMyo dataset provides the necessary data for gesture classification, but it does not include joint angle information.

**Hand Joint Regression Task:** Since the GRABMyo dataset lacks joint angle data, we employed the Ninapro DB2 dataset for this task. Both pre-training and fine-tuning were conducted using data exclusively from Ninapro DB2.

*Participant-Dependent Experiments.* Our experiments are participant-dependent (user-specific). For each user, we train and evaluate the model using data exclusively from that user. This approach aligns with the objectives of our study, focusing on enhancing the model's robustness and performance for individual users without introducing additional data from other

| Task Type | Data Used | Samples per User (except rest) |
|---|---|---|
| **Classification** | | |
| Training Set | Five repetitions per gesture per user | 42,500 samples |
| Validation Set | One repetition randomly selected from the training set | 8,500 samples |
| Test Set | Two repetitions per gesture per user | 17,000 samples |
| **Regression** | | |
| Training Set | Four repetitions per gesture per user | 34,000 samples |
| Validation Set | One repetition randomly selected from the training set | 8,500 samples |
| Test Set | Two repetitions per gesture per user | 17,000 samples |

Table 7: Data Splits for Classification (GRABMyo) and Regression (Ninapro DB2) Tasks

participants during pre-training or fine-tuning. This is a common practice in sEMG studies when aiming to optimize performance for specific users.

## G. Supplementary details of the experimental section

All comparison methods reported in our paper were re-implemented by us. This decision was made because different methods in the literature often use different datasets, select varying subsets of gesture categories, or employ different evaluation protocols, making direct comparisons challenging. By re-implementing these methods under a consistent framework, we ensured that all methods were evaluated:
- On the same dataset(s)
- Using the same set of gesture categories
- Under identical training, validation, and testing conditions
- With consistent hyperparameter tuning strategies based on the original papers

### G.1. Fairness of Comparison Regarding Pretraining

We want to emphasize that the pretraining in our work is fundamentally different from typical pretraining approaches in fields like NLP. Unlike models that are pre-trained on large external datasets and then fine-tuned on specific tasks, our method performs pretraining without introducing any external data. We use only the user-specific training data for both pretraining and fine-tuning. This means that both our method and the comparison methods are trained and tested on exactly the same data, ensuring a fair comparison. Our contribution demonstrates that our approach enhances the model's robustness and performance using the existing user-specific data even without additional data.

### G.2. Evaluation Protocol Compared to the Original Dataset Paper

The original GRABMyo dataset paper did not directly evaluate gesture classification accuracy or provide corresponding benchmarks for classification tasks. Instead, the dataset's quality was assessed using metrics like the Area Under the Curve (AUC) and Equal Error Rate (EER). Therefore, our work does not follow the exact evaluation protocol proposed in the original paper because such a protocol for gesture classification was not established.

### G.3. Details of Evaluation Metrics

**Root Mean Square Error** (RMSE) is a common metric for evaluating the deviation between predicted and observed values. As the range of fluctuations in the curves of different joint angles can vary significantly, it is difficult to fairly evaluate the performance of models using RMSE alone. Normalization of RMSE addresses this issue, resulting in the Normalized RMSE (NRMSE).

$$
\text{RMSE} = \sqrt{\sum_{i=1}^{N} \frac{(\theta_{\text{est}} - \theta_{\text{real}})^2}{N}},
$$
$$
\text{NRMSE} = \frac{\text{RMSE}}{\theta_{\max} - \theta_{\min}}.
$$
(9)

**Pearson Correlation Coefficient** (CC) is a widely used measure of the linear relationship between two variables. It ranges from -1 to 1, where a larger CC value indicates greater similarity between the predicted and estimated joint angles curve,

indicating improved estimation.

$$\text{CC} = \frac{\sum_{i=1}^{N} \left( \theta_{\text{est}} - \overline{\theta_{\text{est}}} \right) \left( \theta_{\text{real}} - \overline{\theta_{\text{real}}} \right)}{\sqrt{\sum_{i=1}^{N} \left( \theta_{\text{est}} - \overline{\theta_{\text{est}}} \right)^2} \sqrt{\sum_{i=1}^{N} \left( \theta_{\text{real}} - \overline{\theta_{\text{real}}} \right)^2}}, \tag{10}$$

where $\theta_{\text{est}}$ and $\overline{\theta_{\text{est}}}$ are the estimated angle and their average, while $\theta_{\text{real}}$ and $\overline{\theta_{\text{real}}}$ are the real angle and their average. $\theta_{\max}$ is the maximum of the real angle, and $\theta_{\min}$ is the minimum of the real angle.

## H. Inference Performance and Parameter Comparison of Models

| Model | Inference Time GPU (A6000) | Inference Time CPU (AMD EPYC 7543) | Parameter Count | GPU Memory Allocated |
|---|---|---|---|---|
| Transformer | 3.8 ms | 15.1 ms | 481169 | 18.08 MB |
| Add **STEM** with weight sharing | 3.9 ms | 17.6 ms | 489233 | 23.66 MB |
| Without weight sharing | 4.8 ms | 27.5 ms | 581137 | 21.65 MB |

Table 8: The comparison of inference time, number of parameters, and GPU usage between the model using **STEM** and a non-weight sharing transformer layer.

In Table 8, we use the following hyperparameters: Feature dimension is 12, the maximum length is 200, model dimension is 64, number of attention heads is 2, number of layers is 3, dimension of feedforward network is 256, number of classes is 17, the dropout rate is 0.1, positional encoding is 'learnable', the activation function is 'GELU', and normalization is 'BatchNorm'. The **STEM** model uses the same parameters as those in the experimental setup of the paper.

**Parameter Count:** By incorporating **STEM**, we significantly reduce the model's parameter count. Specifically, when the **STEM** module is added (with weight sharing enabled), the parameter count increases slightly from 481,169 to 489,233, an increase of about 1.7%. However, if weight-sharing is not used, the parameter count increases substantially to 581,137, a 21% increase.

**Inference Time:** We measured the inference time on both GPU and CPU. When using the **STEM** module with weight sharing, the GPU inference time is 3.9ms, and the CPU inference time is 17.6ms, which is a significant improvement compared to the case without the **STEM** module (4.8ms on GPU and 27.5ms on CPU). This indicates that while enhancing short-term features, the **STEM** module maintains a low inference time.

**GPU Memory Consumption:** The relative increase in GPU usage is due to the mechanism of parallel computation that the GPU activates when using a sliding window.

### H.1. Parameters search and the model's sensitivity

The selection of the masking ratio (0.15) and the average length of masked segments was informed by a systematic hyperparameter search using the (0.05, 0.15, 0.25, 0.35, 0.45), (1,3,5,7,9) range. This was conducted via Wandb, ensuring a comprehensive evaluation of the model's sensitivity to these parameters. We observed that a masking ratio of 0.15 and an average length of 3 consistently yielded the best performance in terms of classification accuracy. While the accuracy fluctuated by approximately 4% across the tested range, the 0.15 ratio provided an optimal balance between introducing sufficient noise for robust feature learning and retaining enough original signal for effective pretraining. The choice of segment length further complements this masking strategy, as it ensures that the model learns to reconstruct meaningful patterns while not overly relying on adjacent unmasked data. These parameters jointly enable the model to focus on capturing intrinsic signal variability, thereby enhancing its resilience to real-world noise scenarios. Such robustness aligns well with our goal of improving generalization in noisy environments.

Regarding window sizes of **STEM**, we experimented with varying configurations and found that a short-term window size of 41 and 21 with a step size of 1 provided the best balance between noise isolation and feature granularity. We use Wandb to search for the best window size; the search space is (11,21,31,41,51,61). The accuracy fluctuated by approximately 3% across the tested range. In the future, we will collaborate with biologists to explore the relationship between window size and different gesture classifications on a biological level.

