# OpenReview forum: "Revisiting Noise Resilience Strategies in Gesture Recognition: Short-Term Enhancement in sEMG Analysis"
_ICML.cc/2025/Conference — ICML 2025 poster_

### Official Review · Reviewer_QgDP · 2025-03-09

**Overall Recommendation:** 4

**Summary:**

This paper proposes a noise-robust method for surface electromyography (sEMG)-based gesture recognition. The authors emphasize the importance of short-term signal learning in mitigating the interference of local noise, which could otherwise degrade the modeling of long-term signals. Specifically, the paper introduces an sEMG module that separately models long-term and short-term signal features. Within this module, the authors propose a advanced masking strategy to prevent excessive signal isolation, a common issue when applying standard masking techniques in masked autoencoders. Additionally, they propose an Asymmetric Optimization method that prioritizes difficult cases during training, enhancing model robustness.The effectiveness of these designs is validated through extensive experiments, demonstrating significant improvements in noise resistance and overall gesture recognition performance.

**Claims And Evidence:**

The validation experiments in Fig. 3 demonstrate the effectiveness of the proposed methods. However, I still have some doubts about whether the reduction of noise interference is directly attributed to the enhancement of short-term feature modeling. While it is a common sense that combining short- and long-term feature modeling generally improves performance, the specific contribution of short-term modeling in mitigating noise interference remains unclear. o strengthen this claim, additional qualitative analysis is needed. For instance, providing insights into the proportion of data affected by local noise and the duration distribution of such noise would help clarify the impact of short-term modeling on noise robustness.

**Essential References Not Discussed:**

I haven't found related papers that were not discussed in this paper.

**Experimental Designs Or Analyses:**

I have reviewed the experimental design, particularly the noise robustness evaluation in Fig. 3. While the results suggest improved performance under different noise conditions, the paper does not provide details on how the noise levels were set or whether they align with real-world conditions. A more rigorous analysis, such as comparing against real sEMG noise distributions, would strengthen the validity of the conclusions.

**Methods And Evaluation Criteria:**

The dataset and evaluation metrics are reasonable in this paper.

**Other Comments Or Suggestions:**

No.

**Other Strengths And Weaknesses:**

1. The paper lacks a direct comparison between employing Focal Loss and Asymmetric Optimization. Evaluating their respective impacts on model performance would provide a clearer justification for the proposed optimization strategy.
2. There is no comparison between conducting long-term and short-term modeling in a non-parallel mode. Additionally, the computational cost differences between the parallel and non-parallel approaches should be supported by experimental results, not just a claim.
3. What types of examples are identified as difficult, and why? Furthermore, what is their proportion in the dataset? Providing more details on this aspect would help clarify the role of Asymmetric Optimization in handling hard cases.
4. Have you considered directly applying the typical masked strategy used in standard reconstruction methods? It would be helpful to provide experimental results comparing this approach with your proposed strategy.
5. What is the proportion of difficult cases in the dataset? Additionally, how does model performance vary across different difficulty levels?

**Questions For Authors:**

No.

**Relation To Broader Scientific Literature:**

[1] A novel event-driven spiking convolutional neural network for electromyography pattern recognition. [2] SpGesture: Source-Free Domain-adaptive sEMG-based Gesture Recognition with Jaccard Attentive Spiking Neural Network haven't been discussed and compared in this paper.

**Theoretical Claims:**

I have reviewed the algorithm for sEMG signal masking and the arguments regarding the advantages of short- and long-term signal modeling. However, the theoretical analysis supporting how short-term modeling improves the model's robustness to noise is currently lacking.

---

> ### Author Rebuttal · Authors · 2025-03-28
>
> # Response to Reviewer QgDP
>
> Thank you for your professional review and valuable time. Your positive assessment is incredibly encouraging to our research team. We sincerely appreciate your thoughtful comments and would like to address each of your concerns as follows:
>
> ## Broader Scientific Literature
>
> Thank you for bringing this to our attention. We will discuss these two critical works (SCN, SpGesture) in the camera-ready version from the perspective of SNN development in sEMG gesture recognition.
>
> ## Focal Loss vs Asymmetric Optimization
>
> Thank you very much for bringing this to our attention. Although this was not our core contribution, we appreciate that providing these results would offer readers more valuable insights.
>
> We have conducted additional tests on the Grabmyo:
> - STET + Focal Loss: 89.42%
> - STET + Asymmetric Optimization: 90.54%
>
> ## Modeling in Non-Parallel vs Parallel
>
> Thank you for your suggestion. We compared the Focal Transformer (non-parallel) with our method.
>
> | Backbone | Grabmyo (ACC) | GPU-A6000-Latency (ms) |
> |----------|---------------|------------------------|
> | Focal Transformer | 84.56 | 4.7 |
> | STET | 90.72 | 3.9 |
>
> In our inference speed experiments, we used a 4-stage Focal Transformer, while STET used two long-term layers plus two short-term layers.
>
> ## Concerns about Hard Samples
>
> We appreciate your insightful question. Below is our detailed response:
> ### Definition of Hard Samples
> **Hard Positives**:
>    - **Definition**: Positive samples (ground-truth $y_{i,j} = 1$) with low predicted probabilities $\widehat{y}_{i,j}$
>    - **Weighting Mechanism**: The term $(1 - \widehat{y}_{i,j} )^{\gamma^+}$ assigns higher weights to samples where the model lacks confidence.
>
> ### Proportions and Dynamics
>
> | Sample Type | Estimated Hard Proportion | Remarks |
> |-------------|---------------------------|---------|
> | **Hard Positives** | 40%-60% (early training) | Decreases to ~20% as the model converges due to asymmetric focusing. |
> | **Hard Negatives** | 5%-10% (persistent) | Suppressed via $\gamma^- < \gamma^+$ and $m=0.2$; absolute count remains high but influence is reduced. |
>
> ## Classic Mask Method vs Our Method
>
> Thank you for your suggestion! We actually included related experiments in Appendix A.3. Classical mask uses a standard pretraining with a general mask, employing a standard masking approach.
>
> | Backbone | Masking method | AG Noise↓ (%) | MG Noise↓ (%) |
> |----------|----------------|--------------|--------------|
> | Transformer | classical | 22.00 | 15.00 |
> | Transformer | ours | 15.00 | 13.00 |
>
> We understand your concern and will highlight this experiment more prominently in the revised paper.
>
>
> ## Concerns about  Noise Reduction
>
> We appreciate your thoughtful comment. We would like to clarify this relationship with additional insights and reasoning.
>
> The connection between short-term feature modeling and noise resilience is rooted in the fundamental characteristics of sEMG signals and common noise patterns.
> 1. **Electrode-skin interface noise**: Movement artifacts and momentary changes in skin-electrode contact typically manifest as brief bursts of interference rather than consistent long-term corruption [1, 2].
> 2. **Environmental electromagnetic interference**: Such interference usually occurs in short, sporadic patterns rather than continuously affecting the entire signal [3].
>
> It's worth noting that we're **not focusing on constant background noise** (such as power line interference at 50/60Hz), as these can be effectively removed through conventional filtering techniques. Our primary concern is with **transient, unpredictable noise patterns** that are more challenging to address through traditional methods and represent common challenges in real-world applications.
>
> By adopting a sliding window attention mechanism, our STEM concentrates on local temporal windows to isolate noise within affected segments, preventing it from spreading across the entire sequence. This method outperforms the long-term decoder under various noise types (as shown in Figure 3) and offers multiple sampling perspectives: overlapping windows that partially include noisy regions can still capture valuable information, thus reinforcing resilience.
>
> **We believe visualizing the noise distribution is a worthwhile idea**; however, precisely pinpointing the noise remains challenging, so we cannot present such a visualization yet, **but we will continue to work toward it**.
>
>
> ### References
> [1] Surface electromyography: physiology, engineering, and applications. IEEE Press.
>
> [2] Filtering the surface EMG signal: Movement artifact and baseline noise contamination. Journal of Biomechanics
>
> [3] Sampling, noise-reduction and amplitude estimation issues in surface electromyography. Journal of Electromyography and Kinesiology
>
> We sincerely appreciate all your valuable suggestions and will incorporate them into our revised manuscript. Your positive assessment is greatly encouraging to our research team.

---

> > ### Comment · Reviewer_QgDP · 2025-04-02
> >
> > I appreciate the authors’ answers to address my concerns, but I still believe that a quantitative analysis of the noise data is essential. Specifically, how many instances of electrode-skin interface noise and environmental electromagnetic interference are present in the dataset? The authors mention the sporadic nature of these artifacts, but this does not replace the need for statistical evidence.
> >
> > If the dataset is large, a reasonable approach would be to use random sampling combined with multi-person manual verification to estimate the prevalence and impact of these noise types. The proportion of noise-contaminated samples plays a significant role in determining this study's overall validity and impact. Without such information, assessing whether the proposed method addresses a widespread problem or only rare edge cases is difficult.
> >
> > Therefore, I will maintain my score until the authors provide a more rigorous quantitative analysis of noise occurrence within the dataset.

---

> > > ### Author Response · Authors · 2025-04-04
> > >
> > > Thank you for your constructive feedback. After detailed research and analysis, our research team has developed an approach to quantitatively analyze the noise distribution in our dataset, as you suggested.
> > >
> > > We have employed two key metrics to analyze the noise distribution quantitatively:
> > >
> > > 1. **Signal-to-Noise Ratio (SNR)**: We analyzed the spectral properties of each sEMG signal recording and measured the SNR (in dB) as the ratio between the power of the signal to the power of the noise. [1] **The power of the noise was estimated as the power of sEMG recordings during the rest trial**  (when sEMG acquisition is least susceptible to interference).[1] The average SNR across all signals in our datasets was 14.565 ±6.385 dB.
> > >
> > > 2. **Correlation Coefficient of Normality (CCN)**: This metric was used to analyze amplitude distribution. For a static contraction with moderate force, sEMG can be modeled as a filtered, random, white Gaussian noise process.[2] It has been suggested that a test of normality can provide a measure of biosignal quality, where a signal amplitude with a non-Gaussian distribution would be considered contaminated. **We generated a Gaussian distribution with equal mean and variance to that of the recording.** [3] The CCN is defined as the Pearson correlation coefficient between the histogram bin values of the sEMG recording and the normal density function value for the corresponding bins.[4] A value close to 1 indicates a normal distribution. The CCN of all signals in our dataset was 0.975±0.041.
> > >
> > > In the real-world dataset,  noises are complexly and randomly intermingled, making precise quantitative isolation impossible. Therefore, we used SNR and CCN calculations to analyze the overall noise distribution in spectral and amplitude properties. Our measurements of the GrabMyo dataset revealed the following distribution:
> > >
> > > | Metric | **High noise (SNR < 10 dB)** | **Moderate noise (10 dB ≤ SNR ≤ 20 dB)** | **Low noise (SNR > 22 dB)** |
> > > | --- | ----------------------------- | ----------------------------------------- | --------------------------- |
> > > | SNR | 28.2% | 56.3% | 15.5% |
> > >
> > > Through visual inspection and based on our experience, we identified samples with **CCN < 0.93** (11.6% of the total samples) as having severe non-Gaussian signals, which may be affected by transient noise such as motion artifacts or electromagnetic interference.
> > >
> > > To demonstrate the effectiveness of our approach across different noise levels [1], we provide comparative model accuracy:
> > >
> > > | Model | **High noise (SNR < 10 dB)** | **Moderate noise (10 dB ≤ SNR ≤ 20 dB)** | **Low noise (SNR > 22 dB)** | **Motion artifacts or EMI (CCN < 0.93)** |
> > > | ------------------- | ----------------------------- | ----------------------------------------- | --------------------------- | ----------------------- |
> > > | Informer | 78.16 | 86.44 | 92.74 | 72.17 |
> > > | Informer+STEM (ours) | 83.32 | 87.28 | 93.13 | 78.63 |
> > > | STET (ours) | 85.93 | 89.62 | 92.89 | 80.22 |
> > >
> > > As the results demonstrate, our method shows significant performance improvements on samples with high and moderate noise levels in real-world datasets.
> > >
> > > We hope this quantitative analysis addresses your concerns regarding the prevalence and impact of noise in real word dataset and demonstrates the practical value of our proposed method. We sincerely thank you for your professional feedback. Your positive assessment is greatly encouraging to our research team.
> > >
> > > [1] Automatic assessment of electromyogram quality. J. Appl. Physiol
> > >
> > > [2] A nonstationary model for the electromyogram. IEEE Trans. Biomed. Eng
> > >
> > > [3] A Review of Techniques for Surface Electromyography Signal Quality Analysis. IEEE Reviews in  Biomed. Eng
> > >
> > > [4] Multi-day dataset of forearm and wrist electromyogram for hand gesture recognition and biometrics.  Scientific Data

---

### Official Review · Reviewer_ybGK · 2025-03-13

**Overall Recommendation:** 3

**Summary:**

This paper specially captures the short-term temporal dependencies in sEMG-based gesture recognition. By designed a self-supervised pretrained method and two short/long-term heads, the proposed method achieve SOTA performance.

## update after rebuttal
The authors have resolved most of my concerns.

**Claims And Evidence:**

Yes

**Essential References Not Discussed:**

N/A

**Experimental Designs Or Analyses:**

Yes

**Methods And Evaluation Criteria:**

Yes

**Other Comments Or Suggestions:**

N/A

**Other Strengths And Weaknesses:**

Pros:
1. This papre is well written and easy to understand with clear visualizations.
2. The proposed methods achieves strong performance.
3. The authors give plentiful ablations and visualizations to support the method.
4. The proposed method seems novel with a new MAE-based self-supervised pre-training method and two heads to capture short/long-term temporal dependencies.

Cons:
1. The methods included for comparison are old. Most are published before 2021 and only one of them is published in 2023. It seems that this area has drawn little attention in recent years, or authors have neglected several recent works.
2. The authors claim that they use two datasets for evaluation (GRABMyo and the Ninapro DB2, line 113), but i only observe results on one dataset (tab.1). It seems that the results are insufficient.
3. While the proposed method seems novel, some of the components have been used in previous methods. For example, MAE-based self-supervised pre-training is widely used in previous self-supervised methods. The long-term modeling method is a simple attention implementations, while the short-term modeling method is implemented by windowed attention.

**Questions For Authors:**

My overall concerns focus on the old methods included for comparison, and whether the authors use adequate datasets for evaluation.

**Relation To Broader Scientific Literature:**

N/A

**Theoretical Claims:**

Yes

---

> ### Author Rebuttal · Authors · 2025-03-27
>
> # Response to Reviewer ybGK
>
> We sincerely appreciate your thorough review and valuable feedback on our manuscript. We are  grateful for your recognition of our paper's strengths, including the **clear writing style**, **effective visualizations**, **strong performance** , **comprehensive ablation studies**, and the **novelty of our approach**. Your positive assessment is greatly encouraging to our research team.
>
> We have carefully considered your concerns and would like to address them as follows:
>
> ## Regarding Dataset Evaluation
>
> We apologize for causing this misunderstanding. While we did indeed evaluate our method on **both GRABMyo (results presented in Table 1) and Ninapro DB2 (results presented in Table 4)**, we acknowledge that this was not sufficiently clear in our presentation. We will revise the manuscript to clearly indicate the comprehensive nature of our evaluation.
>
> Furthermore, to provide even more robust evaluation, we have **added results on an additional dataset, Ninapro DB5**, as shown in the table below:
>
> | Model       | Accuracy (%) |
> |-------------|--------------|
> | LST-EMG-Net | 82.23        |
> | Informer    | 85.22        |
> | TEMGNET     | 80.74        |
> | STET (ours) | **87.61**    |
>
> This enhancement brings our total to **three distinct datasets**, offering a more comprehensive assessment of our method's performance across varied conditions.
>
> ## Regarding Comparison Methods
>
> We thank you for highlighting the need for more recent comparison methods. Following your suggestion, we have **expanded our comparisons to include two state-of-the-art methods published in late 2024**:
>
> | Model                   | GRABMyo (ACC) | DB2 (PCC) | DB5 (ACC) |
> |-------------------------|---------------|-----------|-----------|
> | Spgesture (NeurIPS 2024)| 88.06%        | 0.84      | 86.32     |
> | LRNN (TIST 2024)        | 86.75%        | 0.82      | 86.20     |
> | STET (ours)             | **90.76%**    | **0.88**  | **87.61** |
>
> These additional comparisons demonstrate that **our method maintains its performance advantage even against the most recent approaches** in the field.
>
> ## Our Main Contributions
>
> To summarize, we would like to reiterate the key contributions of our work:
>
> 1. **Introduction of sEMG Signal Masking**: We propose a novel self-supervised pretraining technique using sEMG Signal Masking (Sensor-wise and Contiguous Masked Segments following a Geometric Distribution) to leverage the inherent variability in sEMG data.
>
> 2. **STEM Module for Enhanced Noise Resilience**: From the perspective of improving short-term feature representation, we propose STEM, an adaptive and noise-resistant module. Integrating STEM into various neural networks has demonstrated significant performance gains.
>
> 3. **Improved Noise Resilience**: Extensive experiments confirm that our overall design substantially enhances the noise resilience of models for sEMG data.
>
> We sincerely appreciate your insightful comments, which have helped us improve the quality and comprehensiveness of our manuscript.

---

### Official Review · Reviewer_AUAg · 2025-03-14

**Overall Recommendation:** 3

**Summary:**

The paper addresses the problem of noise resilience in surface electromyography (sEMG)-based gesture recognition. The authors propose a novel Short-Term Enhancement Module (STEM), which focuses on capturing short-term dependencies in sEMG signals to enhance noise resistance. Further, results on GRABMyo and Ninapro DB2 datasets show >20% improvement in noise resilience compared to existing models.

**Claims And Evidence:**

Yes

**Essential References Not Discussed:**

Not to my knowledge.

**Experimental Designs Or Analyses:**

The experiments are missing a robust evaluation based on either Leave one user out or n-fold cross validation to show there is no overfitting in the results.

**Methods And Evaluation Criteria:**

Yes.

**Other Comments Or Suggestions:**

No

**Other Strengths And Weaknesses:**

Strengths:
- The paper is clean and organized.
- Well motivated problem and novel idea of using short-term feature extraction.

Weaknesses:
- Missing robust evaluation of the existing dataset using Leave-one-subject-out (LOSO) evaluation or n-fold cross-validation.
- Since, the gains are small compared to the baselines. It would be good to know the error bar (mean, stddev) of the results and baselines.
- Any evaluation of latency of the system is missing compared to previous works.

**Questions For Authors:**

Please refer to strengths and weaknesses section for the rebuttal.

**Relation To Broader Scientific Literature:**

The paper builds on previous works and advances sEMG-based gesture recognition by focusing on short-term dependencies.

**Theoretical Claims:**

It is an empirical paper.

---

> ### Author Rebuttal · Authors · 2025-03-27
>
> # Response to Reviewer AUAg
>
> We sincerely thank you for your thorough review and **insightful feedback** on our paper. We particularly appreciate your recognition of our **well-motivated problem statement** and the **novel approach of using short-term feature extraction**. Your professional suggestions have been invaluable in improving our work.
>
> ## Regarding N-fold Cross-validation
>
> Thank you for this excellent suggestion. While we initially followed standard evaluation methods in the field of sEMG-based gesture recognition (as seen in [1], [2], [3]) and used validation sets to prevent overfitting (detailed in the appendix's dataset section), we also believe that your suggestion is extremely valuable.
>
> As per your recommendation, we have conducted a **6-fold cross-validation** comparing our method with the second-best performer (Informer):
>
> | Method | Average Accuracy | Standard Deviation |
> |--------|------------------|-------------------|
> | **STET (Ours)** | **92.15%** | **1.12** |
> | Informer | 88.06% | 1.56 |
>
> Statistical significance was confirmed with a t-test (p < 0.001), further supporting the robustness of our approach.
>
> ## Concern about error bars
> We apologize for any confusion. We would like to clarify that we have indeed included the mean and standard deviation in Table 1 of our original manuscript, as well as in our newly added n-fold experiments.
> We appreciate your suggestion and recognize the importance of clearly presenting statistical significance when comparing our approach with the baselines. Based on your feedback, we will revise the manuscript to make these error metrics more prominent and ensure they are easily noticeable throughout our experimental results section.
>
> ## Regarding Inference Latency
>
> We completely agree with your insightful point about the importance of latency evaluation for downstream applications. While this information was included in our appendix under "Inference Performance and Parameter Comparison of Models," we have expanded this section with additional comparative analysis:
>
> | Model | Inference Time by GPU (A6000) | Inference Time by CPU (AMD EPYC 7543) | Average Accuracy |
> |-------|------------------------------|--------------------------------------|-----------------|
> | Transformer (same layer without weight sharing) | 4.8 ms | 27.5 ms | 85.26% |
> | TCN | 2.1 ms | 10.6 ms | 81.50% |
> | GRU | 6.0 ms | 36.0 ms | 86.30% |
> | LST-EMG-NET | 5.2 ms | 31.0 ms | 85.31% |
> | **STEM (Ours)** | **3.9 ms** | **17.6 ms** | **90.76%** |
>
> Our STEM module adds minimal computational overhead (only 0.1ms on GPU and 2ms on CPU) while delivering **superior accuracy with competitive inference speed**.
>
> ## Conclusion
>
> Once again, we are grateful for your expert review and constructive feedback. We hope the additional evaluations address the concerns you raised while further validating the effectiveness of our approach. Your positive assessment is encouraging to our research team.
>
> [1] Multi-attention feature fusion network for accurate estimation of finger kinematics from surface electromyographic signals IEEE Transactions on Human-Machine Systems
>
> [2] Cross-Subject Lifelong Learning for Continuous Estimation from Surface Electromyographic Signal IEEE Transactions on Neural Systems and Rehabilitation Engineering
>
> [3] A CNN-attention network for continuous estimation of finger kinematics from surface electromyography IEEE Robotics and Automation Letters

---

### Decision · Program_Chairs · 2025-05-01

**Decision:**

Accept (poster)

**Comment:**

This work deals with the problem of noise resilience in surface electromyography-based gesture recognition. The authors introduced a new Short-Term Enhancement Module, that focuses on capturing short-term dependencies in sEMG signals to enhance noise resistance. Overall, the method is quite interesting and the idea of using short-term feature extraction is quite novel. All the reviewers recommend acceptance. AC agrees with the reviewers.